

# Network based multifactorial modelling of miRNA-target interactions

Selcen Ari Yuka and Alper Yilmaz

Department of Bioengineering, Yildiz Technical University, Istanbul, Turkey

## ABSTRACT

Competing endogenous RNA (ceRNA) regulations and crosstalk between various types of non-coding RNA in humans is an important and under-explored subject. Several studies have pointed out that an alteration in miRNA:target interaction can result in unexpected changes due to indirect and complex interactions. In this article, we defined a new network-based model that incorporates miRNA:ceRNA interactions with expression values. Our approach calculates network-wide effects of perturbations in the expression level of one or more nodes in the presence or absence of miRNA interaction factors such as seed type, binding energy. We carried out the analysis of large-scale miRNA:target networks from breast cancer patients. Highly perturbing genes identified by our approach coincide with breast cancer-associated genes and miRNAs. Our network-based approach takes the sponge effect into account and helps to unveil the crosstalk between nodes in miRNA:target network. The model has potential to reveal unforeseen regulations that are only evident in the network context. Our tool is scalable and can be plugged in with emerging miRNA effectors such as circRNAs, lncRNAs, and available as R package ceRNAnetsim: https://www.bioconductor.org/packages/release/bioc/html/ceRNAnetsim.html.

## INTRODUCTION

MicroRNAs (miRNAs) are a family of short non-coding RNAs that are key regulators of gene expression through various post-transcriptional mechanisms (*Brennecke et al., 2005*). Although the function of miRNAs is not fully understood, miRNAs predominantly repress their targets. Repressive activities of miRNAs vary depending on many factors that are significant to miRNA:target interactions. These factors include miRNA:target binding energy, binding location in target sequence, base pairing types between miRNA and target, the abundance of miRNAs and targets (*Grimson et al., 2007*). For example, a proteomics study has shown that characteristics of binding, such as seed pairing type and target site location, drastically affect miRNA function (*Xu, Wang & Liu, 2014*). Similarly, another study has revealed that the length of canonical seed base pairing is correlated with the affinity between miRNA and target (*Bosson, Zamudio & Sharp, 2014*). Nucleotide context of miRNA:target complexes determine binding energy (*Cao & Chen, 2012*) and binding energy between miRNA and target indicates stability or affinity of the complex (*Helwak et al., 2013*; *Breda et al., 2015*). Early studies have argued that 2–8 nt

Corresponding author
Selcen Ari Yuka,
selcenay@yildiz.edu.tr

(nucleotide) sequence located in the 5′end of miRNA, known as the seed, binds to a specific sequence located in 3′UTR of its target (*Bartel, 2004*; *Lewis, Burge & Bartel, 2005*). Recent studies have shown that miRNAs can interact with their targets via sequences located in 5′UTR or CDS as well (*Hausser et al., 2013*; *Helwak et al., 2013*; *Moore et al., 2015*). These studies have also shown that binding location either dictates the affinity of miRNA:target interaction or affects level of target degradation. Although initial findings suggested seed sequences are perfectly complementary with their target site (*Bartel, 2009*; *Grimson et al., 2007*), some researchers have reported that seed sequence of miRNA can have mismatches or bulged/wobble nucleotides (*Chi, Hannon & Darnell, 2012*). On top of all these factors, the abundance of miRNAs and targets and miRNA:target ratio in cells predominantly affect the efficiency of miRNA:target interaction (*Arvey et al., 2010*; *Bosson, Zamudio & Sharp, 2014*; *Denzler et al., 2014*).

As it is possible for miRNAs to suppress multiple targets, an individual mRNA can also be targeted by multiple miRNAs. Target mRNAs exhibit competitor behavior for shared miRNAs, hypothesized as competing endogenous RNAs (ceRNAs) (*Ala et al., 2013*; *Cesana & Daley, 2013*). Briefly, the ceRNA hypothesis postulates that alterations of one ceRNA can have notable effects on the ceRNA network since the activity of miRNAs is affected by the abundance of their targets (*Ala et al., 2013*). Regarding the interaction between miRNAs and their targets in a cell, explaining and predicting the aftermath of an individual perturbation is difficult due to the complexity of interactions. Various computational and experimental studies have tackled the problem of unraveling ceRNA: miRNA interactions. For instance, when the abundance of one of the targets of miR-122 increased, the expression levels of remaining targets also slightly increased as a result of decreasing repressive activity of miR-122 on the remaining targets (*Denzler et al., 2014*). A mathematical model has been developed for changes in total target pool concentration after grouping targets according to affinity. It has been demonstrated that miRNA activity correlated with the affinity between miRNA and target (*Bosson, Zamudio & Sharp, 2014*). The cooperative efficiency of miRNAs, as well as competitor behaviors of targets, were also studied and were shown to be crucial for regulating available mRNA levels of targets (*Denzler et al., 2016*). Exploring miRNA:target interactions in the network context was recommended due to the complexity of interactions (*Figliuzzi, Marinari & De Martino, 2013*). Models that can explain miRNA target interactions through topological features were applied to bipartite networks considering direct interactions only (*Nitzan et al., 2014*) or both direct and indirect interactions (*Robinson & Henderson, 2018*). Through common miRNAs and genes, all miRNAs and targets in the network were shown to interact with each other in a bipartite fashion. More recently, an R package called SPONGE was developed to detect computationally valid competing pairs by using the miRNA expression, gene expression, and common miRNAs between gene targets (*List et al., 2019*). Additionally, the miRmapper package utilizes an adjacency matrix to associate miRNAs using differentially expressed genes and identifies significant nodes using topological properties of network (*Da Silveira et al., 2018*).

Previous studies of miRNA:target interactions have not dealt with large-scale networks, comprising of all miRNAs and all their targets in a given organism. In such a network

**Table 1 Summary of networks used in this study.**

| Network name and description | Number of genes/ Expression source | Number of miRNAs/ Expression source | Source of gene:miRNA Interactions | Interaction factors |
|---|---|---|---|---|
| Sample \| Sample Network | 6 Hypothetical | 2 Hypothetical | Hypothetical | None |
| Sample+ \| Sample Network with interaction factors | 6 Hypothetical | 2 Hypothetical | Hypothetical | STE[a]: Hypothetical RE[b]: Hypothetical E[c]: Hypothetical |
| Real \| Large network with experimental expression levels | 3,265 RNA-Seq[d] | 581 miRNA isoform quantification[d] | Predicted (miRTaRBase) and SPONGE | None |
| Real+ \| Large network with experimental expression levels and more acurate pairings | 1,348 RNA-Seq[d] | 284 isoform quantification[d] | Experimental (CLASH, CLEAR-CLiP) and SPONGE | STE: Deduced[e] RE: Deduced[e] E: Calculated[f] |

**Notes:**
[a] Seed Type Effect.
[b] Region Effect.
[c] Energy.
[d] Downloaded from The Cancer Genome Atlas.
[e] Obtained by compiling the variables supported by the experimental datasets into their numerical values.
[f] Retrieved from experimental sources.
Please see Supplemental Materials and Methods for details about data retrieval and network construction.

every node is directly or indirectly connected to any other node through shared miRNAs or common targets. In large-scale miRNA:target network "ripple effect" is expected if a node is perturbed, so expression level changes should be considered in the context of whole network. Emerging high-throughput experimental techniques that pinpoint exact binding locations of miRNAs allow integrating interaction parameters such as binding region and seed sequence. These interaction parameters are required to be integrated for accurate assessment of competition between targets. We developed an approach, not attempted in any existing study, which uses large-scale networks integrating miRNA:target interaction parameters. Addressing the network-wide effect of a perturbation event in a large network has the potential to reveal critical players that cannot be identified by existing methods.

## MATERIALS AND METHODS

### Construction of miRNA:target network

Four separate networks were used throughout our study which either vary in size (Sample vs Real) or lack additional interaction factors (Sample vs Sample+, Real vs Real+), summarized in Table 1 and illustrated in Fig. 1A. For Sample network, arbitrary miRNA:target interactions were generated along with arbitrary expression values. The Sample+ network had additional information, on top of Sample network, regarding miRNA:target interaction (Table 1). miRNA:target pairs from miRTaRBase, expression values of miRNA, and genes from TCGA were fed into SPONGE for sparse partial correlation analysis (List et al., 2019). Resulting miRNA:gene pairs were used to construct the Real network. The Real+ network had additional Cross-linking, Ligation, and Sequencing of Hybrids (CLASH) and Covalent Ligation of Endogenous Argonaute-bound RNAs-Cross-linking Immunoprecipitation (CLEAR-CLiP) (Helwak et al., 2013; Moore et al., 2015) data on

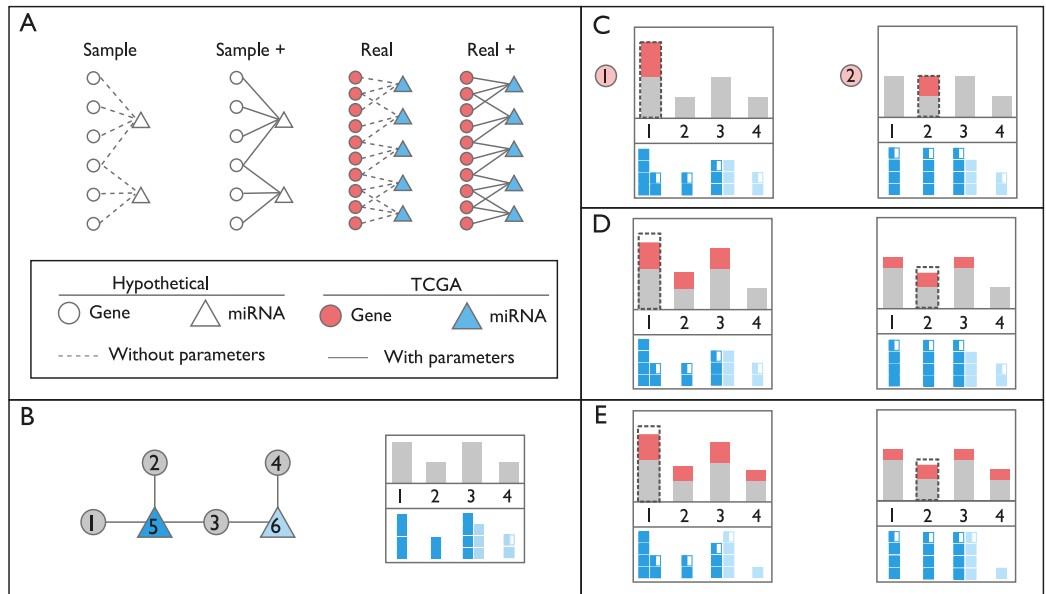

**Figure 1 Schematic representation of networks and perturbation calculations.** (A) Context of the networks used in this study. In hypothetical nodes (empty circle and triangles), arbitrary expression values were assigned. In colored nodes, expression values were retrieved from TCGA. In Sample and Sample+ networks, interactions are arbitrary, in Real and Real+ networks the interactions are either from miRTarBase or experimental results, respectively. Solid lines indicate the presence of additional interaction parameters affecting affinity and degradation. (B) An example of ceRNA network with two miRNAs and four targets. In the box on the right, gray bars on the top panel refer to expression levels of genes. Dark and light blue squares on the bottom panel indicate repression levels of miRNAs on each gene, exerted by miRNAs Node 5 and 6, respectively. Please note that Node 3 is repressed by both miRNAs, Node 5 and 6. (C) Perturbation calculations for two separate triggers by two-fold increase in Gene 1 and Gene 2. The dashed line indicates the initial amount of trigger. The amount of increase in expression of genes is indicated with red bars. miRNA repression levels are recalculated accordingly on bottom panels. (D) Changes in repression levels in (C) are projected on gene expression levels. (E) Due to the change in expression level of the common target gene, Node 3, repression level of miRNA Node 6 on targets, and expression of Node 4 are updated. Perturbation efficiency is calculated by dividing additional expression (red bar) with the original expression level (grey bar) for each gene except the trigger gene (dashed line).

top of Real network. Additional parameters for the Real+ network were curated from literature (see Section 2 in Supplemental Materials and Methods). Complex network analysis and visualization could be performed on constructed networks by tidygraph R package (*Pedersen, 2020*). Moreover, the networks could be exported to Cytoscape by RCy3 package for visualization and further analysis (*Gustavsen et al., 2019*).

## Triggering perturbation and subsequent calculations

In order to demonstrate calculation steps after a perturbation event, Sample network at Table 1 was used. Initially, the network is assumed in steady-state condition (Fig. 2A; Fig. S2) and needs at least one trigger for initiating calculations. The trigger can be a change in the expression level of one or more genes (Fig. 2B; Fig. S3). After the trigger, the network undergoes the iterative cycle of calculations at each of which distribution of miRNA in the local neighborhood is recalculated (Fig. 2C). Based on new miRNA distribution, the expression level of each node (i.e., ceRNA) is updated (Fig. S4). Due to the common
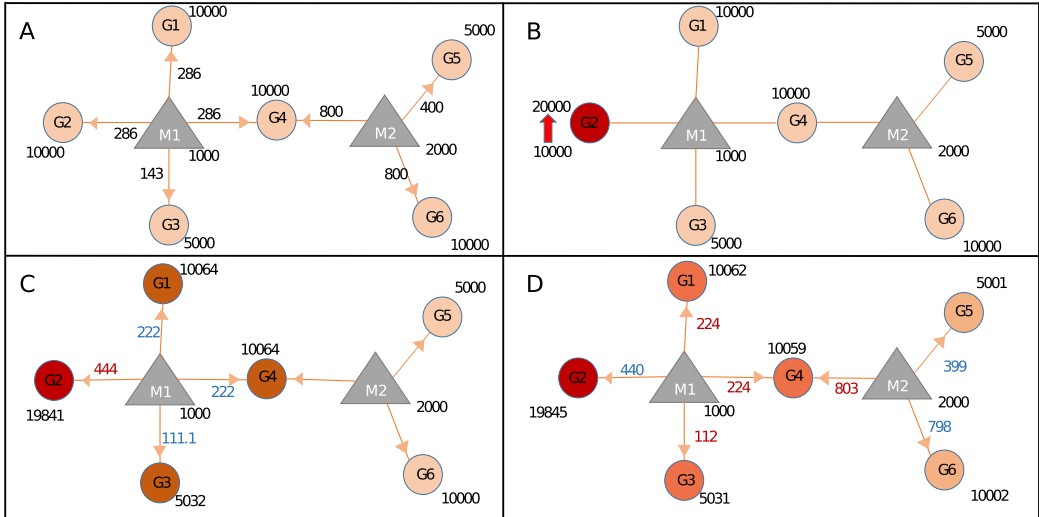

**Figure 2  Calculating changes in gene expression levels with the network-based model.** (A) In steady-state, miRNAs (M: triangles) repress targets (G: circles) according to the proportion of target expression level. Values on edges indicate the initial distribution of miRNAs over their targets. (B) Two-fold increase in transcript level of Gene2 (G2) acts as the trigger (shown with thick red arrow). (C) Distribution of miRNA1 (M1) changes due to increased sequestration of M1–G2 (286–444 units) hence decreasing its distribution to lower levels for G1, G4 (both from 286 to 222), and G3 (from 143 to 111.1), increased or decreased distribution levels are shown in red and blue numbers, respectively. Due to less miRNA targeting, genes G1, G3, and G4 show increased levels of availability, from 10,000 to 10,064 or from 5,000 to 5,032. (D) The change at the expression of common target (i.e., G4) which pulls more M2 hence decreasing availability of M2 for both G5 and G6, consequently levels of G5 and G6 increase due to decreased repression by M2. Expression values are rounded to integers for simplicity. Shades of colors in circles indicate different levels of increase.

targeted nodes, the change in one neighborhood spreads to other neighborhoods (Fig. 2D), consequently has potential to affect the whole network due to the "ripple effect".

During calculations, the following assumptions were adopted; (1) Transcription and degradation rates of miRNAs are steady and equal. (2) All available miRNAs are recycled as in miRNA:ceRNA interactions, targets are degraded and miRNAs are unaffected. (3) ceRNA targets also have stable transcription and degradation rates, and these rates are equal.

The repression level (referred to as Repression Count in formulas) of a miRNA on the individual target ($RC_{ij}$) is calculated according to Eq. (1). Briefly, the expression level of a given miRNA, namely $Exp(miRNA)_j$, is distributed among targets proportional to the expression of an individual target, $Exp(Target)_i$, over total expression levels of targets in the local neighborhood of $miRNA_j$. If an mRNA is targeted by more than one miRNA, the cooperative repression of miRNAs on the target is calculated by summing the repression level of each miRNA (see Fig. 1B, total repression on Node 3, on the right panel).

$$RC_{ij} = Exp(miRNA)_j \times \frac{Exp(Target)_i}{\sum\limits_{i \in \ targets \ of \ miRNA_j} Exp(Target)_i} \qquad (1)$$

## Multifactorial calculations in miRNA:target network

Our model integrates multiple factors when calculating overall miRNA activity.
We classified factors into two categories. Affinity factors determine binding interaction between miRNA and target and they alter the amount of miRNA sequestered to the target. Degradation factors dictate the degradation efficiency of sequestered miRNA on its target. In other words, affinity factors exert their influence before or during binding, degradation factors exert their influence after binding.

From the literature, binding free energy (*Cao & Chen, 2012*; *Helwak et al., 2013*), seed type (*Werfel et al., 2017*) and binding region (e.g., 5′UTR, 3′UTR, CDS) (*Hausser et al., 2013*; *Helwak et al., 2013*) data were retrieved and plugged into Real+ network. For Sample + at Table 1 network we used arbitrary interaction values.

The normalized values of factors are used to determine affinity and degradation of miRNA on its each target (Fig. S5) using Eqs. (2)–(4). Affinity factor of a miRNA:target interaction ($AF_{ij}$) is merely multiplication of normalized energy $E'_{ij}$ with normalized seed type effect $STE'_{ij}$ (Eq. 2). Amount of miRNA per its target ($RC_{ij}$, see Fig. S5C) is calculated by taking affinity factor into account (Eq. 3). Since, not all miRNA:target binding events result in degradation of target, cooperative degradation of an mRNA ($R_i$) by targeting miRNAs is calculated by aggregating multiplication of sequestered miRNA ($RC_{ij}$) with degradation factors such as binding region ($RE'_{ij}$, normalized region effect) (Eq. 4 and Figs. S5D–S5E).

$$AF_{ij} = E'_{ij} \times STE'_{ij} \tag{2}$$

$$RC_{ij} = Exp(miRNA)_j \times \frac{Exp(Target)_i \times AF_{ij}}{\sum\limits_{i \in \ targets \ of \ miRNA_j} (AF_{ij} \times Exp(Target)_i)} \tag{3}$$

$$R_i = \sum\limits_{j \in targetingmiRNAs} RC_{ij} \times RE'_{ij} \tag{4}$$

## Analysis of real breast cancer networks and determining perturbation efficiencies

Expression levels of miRNA and genes in tumor and normal tissues of 87 patients were retrieved from TCGA; https://www.cancer.gov/tcga. We used sparse partial correlation method, SPONGE, to filter out genes with weak association (*List et al., 2019*). Log transformed miRNA and gene expression values were used and genes with $p$-adj less than 0.2 were considered, as suggested by authors (*List et al., 2017*). For Real network, predicted miRNA:target interactions from miRTarBase (*Chou et al., 2017*) and for Real+ network, experimental miRNA:target dataset (i.e., CLEAR-CLIP and CLASH datasets) were used as binary matrix for correlation analysis (*Helwak et al., 2013*; *Moore et al., 2015*).

Our package has functions to calculate perturbed count and perturbation efficiency for each node after using that node as the trigger. The perturbed count is the number of affected nodes, perturbation efficiency is the average of percent change in the expression

level of perturbed nodes. Figures 1B and 1E illustrates perturbation efficiency in a tiny network for two different gene perturbations.

Through the parallel processing, we evaluated the perturbation efficiencies in Real and Real+ networks of all nodes in breast cancer tissue samples (*Tange, 2011*). Throughout the study, we used three-fold increase in expression level as the trigger. Perturbing nodes that show high perturbation efficiency in tumor and normal tissues were subjected to enrichment analysis in KEGG or GO using DAVID (*Huang, Sherman & Lempicki, 2009*; *Huang da, Sherman & Lempicki, 2009*).

## Calculating threshold value for significant perturbation

For a perturbation to be considered significant we calculated the minimum required number of affected nodes and the number of samples (e.g., patient and normal). The distribution of perturbation efficiencies calculated in simulated networks was fitted to finite mixture model (*Trang et al., 2015*). After simulating Real and Real+ networks, we calculated that a node causing perturbation in at least 10 samples with 78 perturbed nodes and nodes that perturbed at least 216 nodes in 18 samples were considered as significantly perturbing, respectively.

# RESULTS

## Basic perturbation calculations using Sample network

To assess the effects of expression level changes in ceRNA regulation based on miRNA and target abundance we constructed the Sample network given in Fig. 2 and Table 1. After an increase in the expression level of a gene (G2), expression values of other genes also changed due to the redistribution of miRNA among its targets. Step by step calculations in gene and miRNA levels were described in Fig. 2. After the increase of expression level of G2, the miRNA that is found in the same neighborhood (M1) became less repressive on its remaining targets (G1, G3, and G4) resulting in increased expression in G1, G3, and G4. The similar finding was shown in an earlier ceRNA hypothesis model (*Ala et al., 2013*). More importantly, expression levels of G5 and G6 changed even though they don't share common miRNA with the initial trigger gene (G2) (Fig. 2D). Previous studies showed that if the abundance of a gene increases remaining targets were affected due to shared targets of multiple miRNAs (*Lai, Wolkenhauer & Vera, 2016*; *Salmena et al., 2011*; *Tay, Rinn & Pandolfi, 2014*). Accordingly, the effect of perturbation in a single gene could spread through the whole miRNA:target network. It's important to note that the changes in gene expression levels will have more pronounced effect if miRNA: target ratio is high which was reported in previous findings (*Arvey et al., 2010*; *Bosson, Zamudio & Sharp, 2014*; *Denzler et al., 2014*).

## Perturbation calculations in Sample+ network: effects of binding energy, seed type and region

Our approach could calculate miRNA repression activity more accurately, by integrating seed type, region, and energy parameters. To demonstrate this capability, we generated Sample+ network, summarized in Table 1 and Table S1. Expression values of Sample and

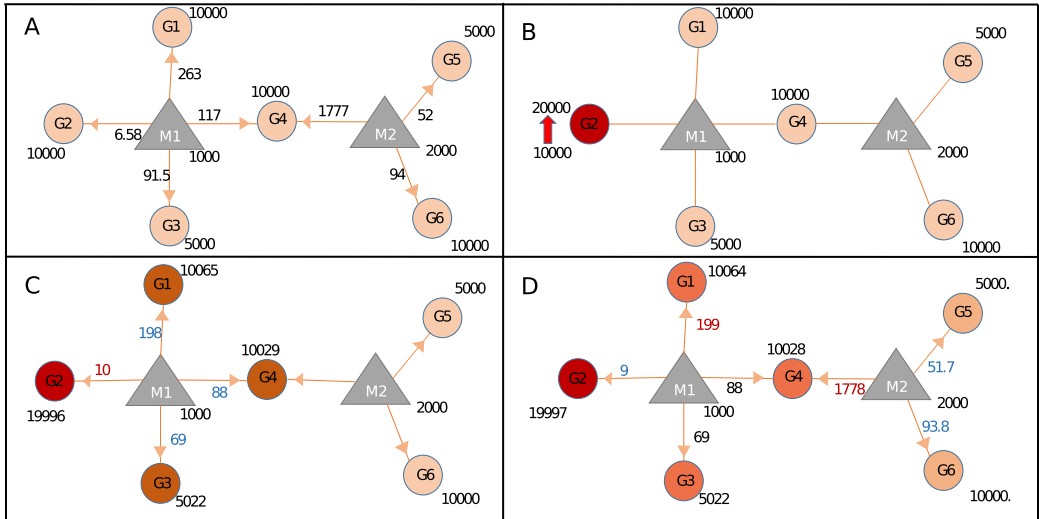

**Figure 3 Target regulations with interaction parameters.** (A) In the steady-state the repression activity of miRNAs on the targets after binding and repression level. (B) The changes the repression activities after increasing of G2 expression. (C) Perturbation of primary neighborhoods of M1 miRNA (M1 miRNA group). (D) Regulation of gene expression of the other gene group via triggering target (common target between M1 and M2).

Sample+ networks were identical however Sample+ contains additional interaction parameters (Table S1) because of which steady-state miRNA distribution in Sample+ (Fig. 3A; Fig. S6) was different than that of Sample network (Fig. 2A). For instance, proportional distribution of G1:M1 interaction in Fig. 2A was higher than the G1:M1 interaction in Sample+ network (Fig. 3A; Fig. S5C) (286 vs 6.58). When the expression of Gene2 (G2) increased (Fig. 3B; Fig. S7), expression values of all genes also changed at various levels because of the contribution of interaction parameters (Figs. 3B–3D; Figs. S7 and S8).

In Sample+ network, G4 was targeted by two miRNAs, M1 and M2, hence expression level changes in G4 have more prominent effect in the whole network when compared to G2 (Fig. S9). The fact that each node had different expression levels and topological features suggested each node had different perturbation efficiency. Our package could screen all nodes for their perturbation efficiencies. When Sample and Sample+ networks were screened for perturbation efficiencies, the results were drastically different (Table S6 and Section 1 in Supplemental Materials and Methods) although Sample and Sample+ networks had the same expression values. This finding suggested additional parameters for miRNA:target interactions should be integrated if available for more accurate calculations.

## Compiling realistic network and identifying nodes causing widespread perturbation

For large-scale analysis, we constructed the Real network with miRNA:target pairs from miRTarBase comprising of 3,265 genes and 581 miRNAs (Table 1). Many instances of the Real network were constructed by overlaying the expression data of each patient. We constructed 174 networks from 87 patients who had both normal and cancer

expression data available. Subsequently, the perturbation efficiency of each node in all networks was calculated. The finite mixture model was used to find significantly perturbing nodes as described in "Materials and Methods". As a result, 70 of 3,265 genes and 27 of 581 miRNAs were found to have significant perturbations in both tissues (Fig. 4A). It was observed that 29 genes and 1 miRNA had tumor tissue-specific perturbing activity. Additionally, in normal tissue samples of these 87 patients, 46 genes and 4 miRNA had showed robust perturbation efficiency. Please note that not all tumor-specific perturbing genes exhibit differential expression between normal and cancer tissues (shown in Fig. S11). Thus, our approach had potential to spot a gene that has insignificant change in expression but becomes effective in relation to expression levels of remaining one or more nodes in miRNA:target network.

In order to assess the functional annotation of highly perturbing genes, enrichment and disease associations were examined. The considerable number of these genes, exactly 55, were enriched ($p$-value < 0.05) in critical pathways in cancer such as PI3K-Akt signaling pathway (4.69 fold enrichment, $p$-value 2.04e−08), proteoglycans in cancer (6.07 fold, $p$-value 1.03e−07), FoxO signaling pathway (6.64 fold, $p$-value 4.55e−06). It was observed that 126 of these genes had enriched in biological processes and molecular functions, including negative regulation of apoptotic process (GO:0043066, 4.75 fold, and $p$-value 5.15e−07), cellular response to epidermal growth factor stimulus (GO:0071364, 23.12 fold, and $p$-value 5.33e−06), cadherin binding involved in cell-cell adhesion (GO:0098641, 8.25 fold, $p$-value 1.35e−11), transcription factor binding (GO:0008134, 5.32 fold, $p$-value 1.54e−05). The full list of enriched GO terms and KEGG pathways is presented in Table S7. Interestingly, tumor-specific perturbing genes showed enrichment in cancer-associated pathways and biological processes (Fig. 4B) while normal tissue-specific perturbing genes were not enriched significantly in the same pathways or processes. While four of the top five enriched KEGG pathways were evidently associated with cancer, the remaining pathway (focal adhesion) was shown to be associated with adhesion, migration, and invasion in breast cancer (*Luo & Guan, 2010*). Additionally, tumor-specific perturbing genes were enriched in several GO terms such as regulation of amino acid metabolism and apoptotic processes which play a significant role in tumor progression (Fig. 4B) (*Vettore, Westbrook & Tennant, 2019*). Also, the negative regulation of transcription regulation from RNA polymerase II (GO:0000122) may be important in manipulating tumor microenvironment and communications of cancer cells (*Venkatraman et al., 2020*). In utero embryonic development process (GO:0001701) plays role in breast cancer metastasis and mammary development in utero (*Howard & Veltmaat, 2013*). The full list of GO terms and KEGG pathways enriched by tumor-only perturbing genes is presented in Table S8.

DisGeNet platform (*Piñero et al., 2020*), a database listing critical genes for numerous diseases, was used to analyze critical genes in our findings. We found that 29 of 99 perturbing genes in tumor tissue had breast cancer disease association score greater than 0.1, with 6.95e−08 hypergeometric $p$-value and 2.9 fold enrichment.

In addition to highly perturbing genes, our analysis revealed highly perturbing miRNAs. In order to associate highly perturbing miRNAs with diseases, The Human microRNA

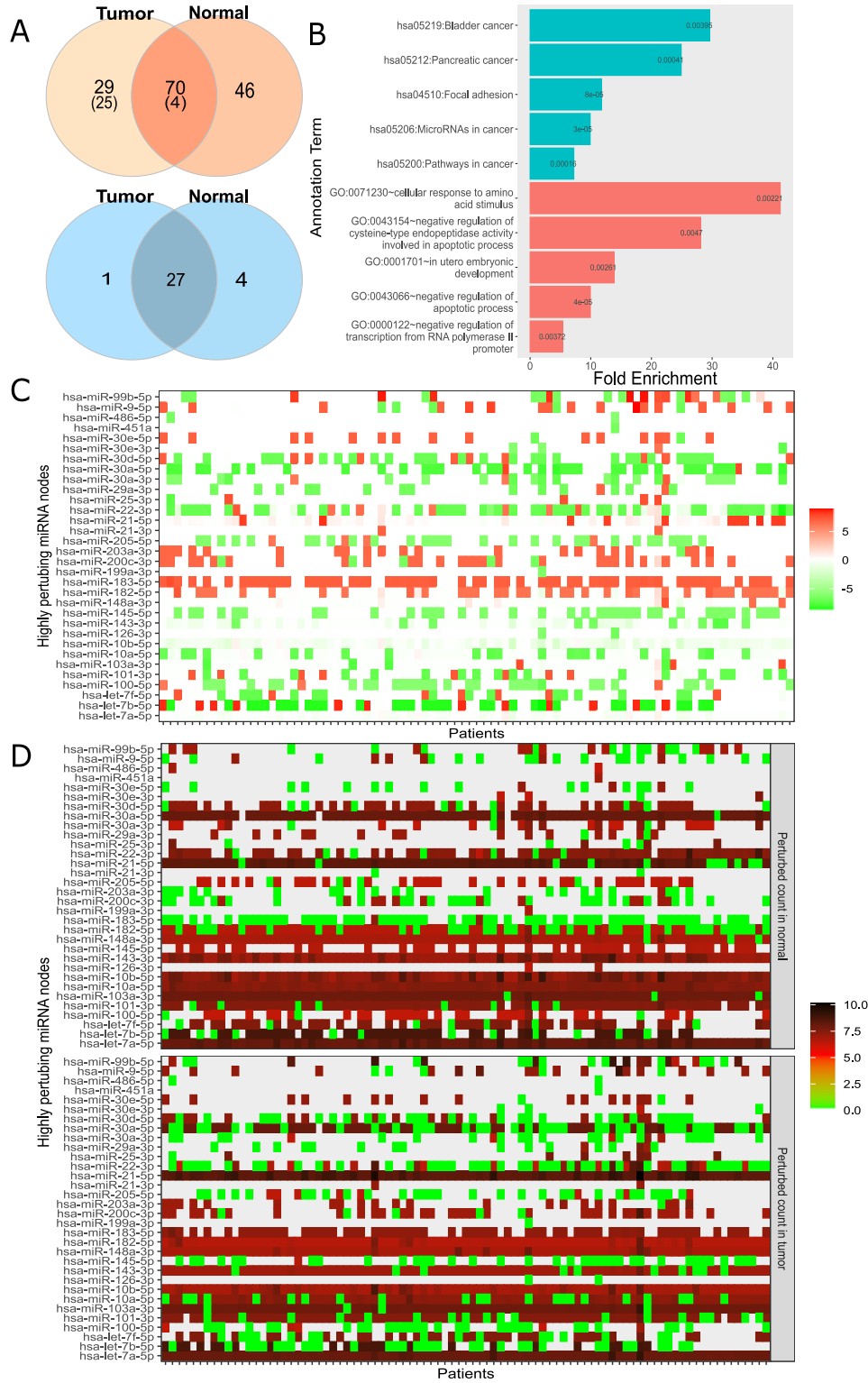

**Figure 4 Analysis of highly perturbing genes and perturbed node counts for all miRNAs in real network.** (A) Number of highly perturbing nodes (genes in orange, miRNAs in blue) with miRNA:gene target pairs, 99 perturbing genes were analyzed at DisGeNet. Number of breast cancer associated genes are indicated in parenthesis. DisGeNet analysis was not performed for normal specific perturbing genes. (B) Top five enriched KEGG pathway (teal) and Gene Ontology (red) terms using all

**Figure 4** (continued)
tumor-specific highly perturbing genes, (C) Comparison of perturbed node counts in tumor and normal tissues for each miRNA. Represented as log2 of affected node count ratio between tumor and normal samples, (D) Affected node number of critical miRNAs in tumor and normal tissues. Represented as log2 (affected node count).

Disease Database (HMDD, Version 3.2) was used (*Huang et al., 2018*). Almost all highly perturbing miRNAs (31 of 32, Fig. 4A) in our study were denoted as breast cancer-associated (2.42 fold enrichment with *p*-value 7.12e−11) in HMDD, 20 of which were attributed with causality. The remaining non-breast cancer-associated miRNA, miR-99b, may play a regulatory role in proliferation and migration processes in breast cancer by affecting the TGF-β signaling pathway (*Turcatel et al., 2012*).

It's worth mentioning that the analysis thus far evaluated 87 normal and tumor samples as a whole. However, per sample basis analysis is also important since miRNAs or genes may show unanticipated behavior at the individual sample level. Figures 4C and 4D summarize perturbed node counts for each miRNA across all samples. For example, miR-30a-5p isoform which suppresses proliferation, migration, and tumor growth (*Yang et al., 2017*) had been observed as highly effective in almost all normal tissue samples but not in all tumor tissue samples. As another example, miR-183 which was commonly up-regulated in tumor samples and one of the key regulators of the metastatic process in breast cancer tissues (*Cao et al., 2020*), exhibited diverse perturbing efficiencies between normal and tumor tissues in our analysis. miR-182-5p and miR-21-5p had been found as highly efficient in tumor tissues, but not in all normal tissues. These results suggested that our ceRNA network-dependent approach can process and interpret many individual networks.

## Detecting perturbation efficiency of nodes in presence of interaction parameters

For more accurate analysis we generated a network with experimental data. The interactions in the Real+ network was originated from CLEAR-CLiP and CLASH datasets as opposed to the Real network which was comprised of predicted miRNA:target interactions. A total of 1,348 genes and associated 284 miRNAs were compiled as the Real+ network (Table 1; Fig. 1A). While significant factors such as energy, seed type, and region on the target sequence in miRNA:target interaction in the template network were the same for each patient data set, expression levels of miRNA and gene were different in each data set. We calculated the perturbation efficiencies for each node in each patient.

By fitting distribution of node perturbations of simulated networks to finite mixture model, a node was defined as a "perturbing node" if it affected at least 216 nodes in at least 18 samples. It had been observed that 475 of 1,348 genes showed high perturbation efficiency in normal and tumor tissue samples. While 12 of these genes were highly effective in tumor tissues specifically, no genes were detected showing normal tissue-specific effect. On the other hand, 83 of 284 miRNAs were found to be a highly

perturbing node in both tissues. KEGG or GO term enrichment analysis of 475 perturbing genes by DAVID tool (*Huang, Sherman & Lempicki, 2009*; *Huang da, Sherman & Lempicki, 2009*) (Fig. S10) showed that 60 of these genes were enriched in signal pathways like FoxO signaling pathway (*p*-value 0.0058 and 2.78 fold enrichment). Also, 399 of perturbing genes were significantly enriched in cancer-associated biological process and molecular functions such as cadherin binding involved in cell-cell adhesion (GO:0098641 with *p*-value 1.19e−08 and 3.68 fold enrichment), cell-cell adhesion (GO:0098609 with *p*-value 5.30e−08 and 3.65 fold enrichment), microtubule cytoskeleton organization (GO:0000226 with 5.23e−04 *p*-value and 4.82 fold enrichment), negative regulation of Ras protein signal transduction (GO:0046580 with *p*-value 3.80e−03 and 7.61 fold enrichment). The full list of enriched GO terms and KEGG pathways is presented in Table S9. When we analyzed the critical genes associated with breast cancer from the DisGeNET database (*Piñero et al., 2020*), 32 of the disease-related genes with a score greater than 0.1 were observed (1.50 fold enrichment with *p*-value 0.0051).

HMDD database analysis showed that 67 of 79 miRNAs (2.13 fold enrichment with p-value 2.05e−17) had been associated with breast cancer and 40 of these miRNAs reported as having the potential to cause breast cancer (*Huang et al., 2018*). The remaining 12 miRNAs not associated with breast cancer in HMDD were explored further in the literature. There are numerous studies that associate breast cancer with miR-28, miR-1287, miR-3065, miR-500a, and miR-99b (*Yang et al., 2011*; *Schwarzenbacher et al., 2019*, *Martinez-Gutierrez et al., 2020*; *Aushev et al., 2016*; *Turcatel et al., 2012*). They have the potential to be used for biomarker, therapeutic or regulatory purposes. Additionally, miR-501, miR-532, and miR-589-3p, which had perturbation efficiency in tumor tissue, were not associated with breast cancer in HMDD. Several researches about miR-589-5p association with drug response and abnormal regulation of miR-532-5p in breast cancer were reported (*Uhr et al., 2019*; *Huang et al., 2020*). miR-577 had been reported as a potential target for breast cancer therapy in an earlier study due to its function as a suppressor of metastasis that is induced by epithelial-mesenchymal transition (*Yin et al., 2018*). On the other hand, at the time of writing this study, there were no reports that directly associate miR-2116, miR-3127, and miR-501-3p with breast cancer.

Compared to simulations made with networks constructed with miRTarBase data set, the Real network, results from Real+ network showed that the high number of genes/miRNAs perturbed the large number of nodes (*Chou et al., 2017*). This finding emphasized the importance of integrating miRNA:target interaction parameters as they profoundly affect the proportional distribution of miRNA expression. As more miRNA:gene interaction parameters become available in future studies by novel experimental methods, our method has the potential to integrate them and thus providing more accurate depiction of ceRNA network perturbations and their consequences.

## Evaluation of performance and runtime
Since human miRNA:target networks comprise thousands of nodes and interaction we evaluated the runtime of our package. To benchmark runtime, 500, 1,000, 5,000, and

10,000 miRNA:target pairs were sampled from the network constructed using miRTarBase data set. Using these networks, the runtime results of simulations originating from a single node and perturbation efficiency analysis of all nodes in networks were benchmarked. It had been determined that runtime increased significantly with increasing network size, for instance, perturbation efficiency analysis for 1,000 and 10,000 interactions takes 84 and 245 s, respectively (Figs. S13 and S14).

## DISCUSSION

Compared to earlier attempts of analyzing miRNA:target interactions, our approach handles large-scale networks while integrating interaction parameters on top of expression level data. Although kinetic modeling is more appropriate and accurate for modeling expression levels, its calculation for large networks is impractical. Thus we used a steady-state approach to process large networks, in other words, transcription, degradation or binding rates of miRNAs, or mRNAs were not considered during calculations. Additionally, other regulation parameters such as gene-gene interactions and activations by transcription factors were ignored as well. Such simplifications allowed calculations in large-scale networks.

Although we considered only mRNAs as ceRNAs in our approach, other non-coding RNA (ncRNA) types are known to compete for miRNA interaction, consequently acting as sponges for miRNAs. Such ncRNAs play role in the regulation of crucial genes. For instance, critical genes extracted from a study involving lncRNA-miRNA-mRNA interactions overlapped with genes important for cancer (*Kesimoglu & Bozdag, 2020*). More specifically, the involvement of lncRNA-miRNA-mRNA interactions with breast cancer and the role of circRNA-miRNA-mRNA interactions in cervical cancer were shown in recent studies (*Tuersong et al., 2019*; *Wang et al., 2020*; *Gong et al., 2019*). As more ncRNA expression data become available our calculations can accommodate them in a large-scale network.

Our approach is not only capable of integrating various players, but also various parameters. Yet to be discovered interaction parameters affecting the binding and function of miRNAs can easily be integrated into calculations. For instance, recent findings show that not all miRNA:target binding events result in functional repression (*Liu & Wang, 2019*). As soon as its data becomes available such a parameter can be included in our calculations as an additional multiplier to Eqs. (2) and/or (4).

The main goal of our approach is not accurately predicting expression levels, which is done by kinetic models, although in very small networks. Our method aims to pinpoint critical players in a large miRNA:target network by integrating diverse players and various parameters. The objective of the proposed approach is to address the network-wide effect of a perturbation event in a large network. Moreover, it enables revealing nodes whose change likely to cause extensive changes in the network.

Differential expression analysis is commonly used for identifying genes or miRNAs that are important between two conditions, usually normal vs disease. Our approach has the potential to reveal critical genes even though their expression does not change, for

instance, some of the tumor-specific perturbing genes in Real network had comparable expression levels between normal and tumor samples (Fig. S11). By harnessing the complexity of the ceRNA network, we observed that a gene or miRNA might become critical if the expression level of the gene(s) in its local neighborhood changes. As an example, miR-30a-5p had comparable expression between normal and tumor samples (Fig. S12), however, it showed high perturbation count in the normal sample only (Figs. 4C and 4D). Considering the fact that miR-30a-5p suppresses proliferation, migration, and tumor growth (*Yang et al., 2017*), our finding might explain how miR-30a-5p loses its suppression without changing its expression level.

In small datasets, screening each node for perturbation efficiency have relatively short runtimes (*Yuka & Yilmaz, 2020*). However, large networks integrating many miRNA: target interactions require the massive amount of calculations which increases the runtime significantly. Although we took advantage of the parallel processing capabilities of various R packages, the runtime for perturbation efficiency calculation was still longer than anticipated.

## CONCLUSIONS

We developed an R package that has the ability to integrate expression values of mRNA and miRNAs along with their interaction parameters into a large miRNA:target network and subsequently calculate the consequences of a perturbation. Taking the competition between mRNAs into account provides more accurate analysis of ceRNA networks. The ceRNAnetsim package is extensible by integrating additional interaction parameters as more experimental data become available. Moreover, additional players in the ceRNA network such as non-coding RNAs can be integrated into calculations as soon as their expression levels and miRNA targeting data becomes available. Our package is able to reveal critical genes that are not discovered by conventional approaches such as differential gene expression analysis. Consequently, our package may help researchers tackle complex interactions in ceRNA networks with a novel approach, leading to better understanding and predictions of abnormal regulations and pathways underlying diseases or conditions.

## ACKNOWLEDGEMENTS

The numerical calculations reported in this paper were partially performed at TUBITAK ULAKBIM, High Performance and Grid Computing Center (TRUBA resources).

### Funding

The authors received no funding for this work.

### Competing Interests

The authors declare that they have no competing interests.

## Author Contributions

- Selcen Ari Yuka conceived and designed the experiments, performed the experiments, analyzed the data, prepared figures and/or tables, authored or reviewed drafts of the paper, and approved the final draft.
- Alper Yilmaz conceived and designed the experiments, performed the experiments, analyzed the data, authored or reviewed drafts of the paper, and approved the final draft.

## Data Availability

Codes and basic method used in study are available at Bioconductor/ceRNAnetsim as an R package: https://www.bioconductor.org/packages/release/bioc/html/ceRNAnetsim.html

Code and data are also available at GitHub: https://github.com/selcenari/ceRNAnetsim, https://github.com/selcenari/ceRNA_multifactorial_model.

## Supplemental Information

Supplemental information for this article can be found online at http://dx.doi.org/10.7717/peerj.11121#supplemental-information.

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
