# Peer review of "Network based multifactorial modelling of miRNA-target interactions"

_PeerJ, doi:10.7717/peerj.11121_

## Round 0.1 · original submission · Major Revisions

Based on the reviewers' comments, we would suggest more clear writing and workflow to explain those details like the default cutoff to narrow down the network size.

Reviewer 1 ·

Basic reporting

Based on ceRNA, this article comprehensively considered the influencing factors of the
interaction between miRNA and mRNA, and builds a model to discover perturbing genes
and miRNAs, which helps to understand the interaction between miRNA-mRNA.

In the preparation process of the article, there are the following suggestions.
1. In the results and discussion part, the content covered in this article should be in the past tense or past perfect tense, rather than the present perfect tense which the author uses more. The content described in the cited literature is the general present tense.
2. The use of the cited documents in the article is not accurate, please refer to the writing of other articles.
3. Part of the expression was difficult to understand.
Line 243-244

Experimental design

The research ideas of the article are clear.
But the author can still consider drawing a flowchart. For bioinformatics analysis research reports, the flowchart can help readers quickly and accurately understand the research ideas of the article. At the same time, it can also be used as a supplement to Table 1, so that readers can effectively identify the difference between Sample/Sample+ and Real/Real+.

Validity of the findings

1. For Figure 11S, it would be better to label the significant difference in gene expression levels between normal tissues and cancer tissues.
2.The article elaborated on the research results that he considered important, but failed to display the relevant data results completely and clearly.
Line 274-285, Line 219-226.

Additional comments

Authors can refer to similar research articles to improve the writing skills of research papers.

Annotated reviews are not available for download in order to protect the identity of reviewers who chose to remain anonymous.

Reviewer 2 ·

Basic reporting

no comment

Experimental design

no comment

Validity of the findings

no comment

Additional comments

Selcen Ari Yuka and Alper Yilmaz present a useful bioinformatics tool to run ceRNA analysis. The introduction provides a good, generalized background of the topic that quickly gives the reader an appreciation of the wide range of applications for this tool. The online R document is neat and easy to follow.
I have a few comments as below:
1, since one miRNA might turn off thousands of coding genes, the ceRNA network is very huge. What is the cutoff to control the size of the constructed ceRNA network.
2, Your example analysis is just focusing on one TCGA case. I am wondering how to conduct multiple samples-based ceRNA network construction? Shall we just choosed concordant regulatory motifs? It will be better to discuss this.
3, The network visualization in R might not the best choice. So, I am wondering if the network file could be imported in Cytoscape for further analysis.
4, it is not very clear about what is the “perturbation efficiency”, and how this could be applied to the ceRNA network construction.

·

Basic reporting

- This manuscript has many small grammer issues that should be fixed prior to publication.

Experimental design

- Runtime analysis: The authors comment on this but do not make any concrete statements, e.g. "perturbation efficiency calculation is still longer than anticipated" or "Our tool can successfully simulate perturbations in large network" which do not give an impression of the limitations of the tool.
- In Equation (1), I still don't understand why i is outside the sum since it is also defined as a running variable. I presume this is the relative expression of one gene i vs all other genes (say gene j). Correct me if I'm wrong, but this would just be the read count normalized by sequencing depth of the RNA-seq experiment, i.e. counts per million or CPM.
- The methods section does not explain how many iterations of the method are run and if there is an automated stopping criterion for determining the steady state.

Validity of the findings

- In the discussion, the authors higlight that a dynamic simulation considering degradation rates etc. would be impractical but what's missing is a statement why the results of ceRNAnetsim are still of importance, e.g. the results do not capture these important factors but nevertheless help to understand the network dynamics in ceRNA regulation and pinpoint critical players that may be worth studying in a follow-up.

Additional comments

The competing endogenous RNA effect is an important concept for understanding systemic effects of microRNA regulation in which we consider miRNAs as resource various transcripts with binding sites compete over. ceRNA regulation can be studied on a global (genome-wide) or on a local scale using partial correlation or conditional mutual information based measures. However, we lack a tool allowing us to study the effect of expression changes on the ceRNA network. The authors close this gap and offer an R package that simulates miRNA target repression in a ceRNA networks over a large number of iterations. This allows studying indirect effects and to assess the potential of single genes or miRNAs to disturb this complex networks. The simulations that can be created here are likely of limited practical use as important mechanisms such as degradation are not considered. Nevertheless, I find the user-friendly, openly accessible and well documented R package provided here useful for the community for gaining a better understanding of the ceRNA theoretical aspects.

I've previously reviewed this manuscript and find many of my previous comments successfully addressed by the authors. Most importantly, the authors conducted additional analyses I requested that leverage information from more than one patient and use ceRNA network inference methodology to refine the network.

---

## Round 0.2 · Minor Revisions

I can see the authors improved the manuscript by adding more details. However, please address some concerns from reviewers directly, like how to export to Cytoscape, which R function should be used, or any reusable code for a similar job. Same for the questions about multiple samples, please share some workable example or code for reviewers.

---

## Round 0.3 · accepted · Accept

Please follow publication instructions to update your manuscript.